# Novel DNA Markers for Identification of *Actinobacillus pleuropneumoniae*

Gun Srijuntongsiri,a Atiwat Mhoowai,b Sukuma Samngamnim,c Pornchalit Assavacheep,c ⓘ Janine T. Bossé,d ⓘ Paul R. Langford,d Navaporn Posayapisit,b Ubolsree Leartsakulpanich,b ⓘ Warangkhana Songsungthongb

aSchool of Information, Computer, and Communication Technology (ICT), Sirindhorn International Institute of Technology, Thammasat University, Pathum Thani, Thailand
bNational Center for Genetic Engineering and Biotechnology (BIOTEC), National Science and Technology Agency (NSTDA), Pathum Thani, Thailand
cDepartment of Veterinary Medicine, Faculty of Veterinary Science, Chulalongkorn University, Bangkok, Thailand
dSection of Paediatric Infectious Disease, Department of Infectious Disease, Imperial College London, London, United Kingdom

**ABSTRACT** *Actinobacillus pleuropneumoniae* causes porcine pleuropneumonia, an important disease in the pig industry. Accurate and sensitive diagnostics such as DNA-based diagnostics are essential for preventing or responding to an outbreak. The specificity of DNA-based diagnostics depends on species-specific markers. Previously, an insertion element was found within an *A. pleuropneumoniae*-specific gene commonly used for *A. pleuropneumoniae* detection, prompting the need for additional species-specific markers. Herein, 12 marker candidates highly conserved (99 – 100% identity) among 34 *A. pleuropneumoniae* genomes (covering 13 serovars) were identified to be *A. pleuropneumoniae*-specific *in silico*, as these sequences are distinct from 30 genomes of 13 other *Actinobacillus* and problematic [*Actinobacillus*] species and more than 1700 genomes of other bacteria in the *Pasteurellaceae* family. Five marker candidates are within the *apxIVA* gene, a known *A. pleuropneumoniae*-specific gene, validating our *in silico* marker discovery method. Seven other *A. pleuropneumoniae*-specific marker candidates within the *eamA*, *nusG*, *sppA*, *xerD*, *ybbN*, *ycfL,* and *ychJ* genes were validated by polymerase chain reaction (PCR) to be specific to 129 isolates of *A. pleuropneumoniae* (covering all 19 serovars), but not to four closely related *Actinobacillus* species, four [*Actinobacillus*] species, or seven other bacterial species. This is the first study to identify *A. pleuropneumoniae*-specific markers through genome mining. Seven novel *A. pleuropneumoniae*-specific DNA markers were identified by a combination of *in silico* and molecular methods and can serve as additional or alternative targets for *A. pleuropneumoniae* diagnostics, potentially leading to better control of the disease.

**IMPORTANCE** Species-specific markers are crucial for infectious disease diagnostics. Mutations within a marker sequence can lead to false-negative results, inappropriate treatment, and economic loss. The availability of several species-specific markers is therefore desirable. In this study, 12 DNA markers specific to *A. pleuropneumoniae*, a pig pathogen, were simultaneously identified. Five marker candidates are within a known *A. pleuropneumoniae*-specific gene. Seven novel markers can be used as additional targets in DNA-based diagnostics, which in turn can expedite disease diagnosis, assist farm management, and lead to better animal health and food security. The marker discovery strategy outlined herein requires less time, effort, and cost, and results in more markers compared with conventional methods. Identification of species-specific markers of other pathogens and corresponding infectious disease diagnostics are possible, conceivably improving health care and the economy.

Address correspondence to Warangkhana Songsungthong, warangkhana.son@biotec.or.th.

The authors declare no conflict of interest.

**KEYWORDS** species-specific DNA markers, *Actinobacillus pleuropneumoniae*, porcine pleuropneumonia, diagnostics, marker discovery

Porcine pleuropneumonia is an important disease with high economic impact for the swine industry (1, 2). Economic loss from the disease is attributed to pig mortality, reduction in daily weight gain, a longer rearing period, lower feed efficiency, as well as medication and veterinary expenses (1, 2). Porcine pleuropneumonia affects pigs of all ages. The disease can be acute with fibrino-hemorrhagic and necrotizing pneumonia, leading to sudden death (3, 4). Pigs that survive acute infection or recover after remedial treatment may become disease carriers (3, 4). It is therefore important to monitor pigs for pleuropneumonia to ensure that they remain free of the disease to promote animal health, food security, and the economy.

The causative agent of porcine pleuropneumonia is *Actinobacillus pleuropneumoniae*, a Gram-negative bacterial pathogen of the pig respiratory tract. This species currently consists of 19 serovars (5), which can be distinguished mainly by unique capsular polysaccharide (CPS) antigens, as lipopolysaccharide O-antigens (LPS O-Ags) can be shared by groups of serovars such as 1/9/11, 3/6/8/15 and 4/7 (6, 7). Despite some genomic differences among various serovars, core genes exist (8) and potentially contain species-specific DNA markers.

*A. pleuropneumoniae* diagnostics are important for surveillance, prevention, and control of porcine pleuropneumonia. Effective diagnostics can guide decisions on antibiotic treatment, quarantine, and vaccine usage. Diagnosis based on clinical signs can be unreliable, as symptoms may be common to various respiratory diseases. The ability to correctly identify and distinguish the species of interest from closely related species is important for guiding an appropriate response to a disease outbreak. DNA-based detection methods such as polymerase chain reaction (PCR) can be highly specific, allowing discrimination of different species when the targeted DNA sequences are sufficiently unique. Amplification of *A. pleuropneumoniae*-specific DNA in pig-derived samples (e.g., lung tissues, nasal swabs, tonsils, and oral fluids) is therefore exploited for disease diagnosis (9–12).

Many DNA markers and PCR assays for *A. pleuropneumoniae* detection have been reported (3, 5, 9, 13–15). Some assays, however, have limitations regarding their specificity, as they are unable to distinguish *A. pleuropneumoniae* from closely related *Actinobacillus* species (3, 5, 13–16). Assays based on the *apxIVA* gene, encoding a repeats-in-toxin (RTX) family protein, are *A. pleuropneumoniae*-specific (9), making this gene an excellent target for *A. pleuropneumoniae* detection. However, mutations within species-specific markers, especially at or within primer binding sites, can lead to diagnostic evasion (5, 17–19). An example of serodiagnostic escape in *A. pleuropneumoniae* is the AP76 strain which contains the IS*Apl1* insertion element in the *apxIVA* gene. The insertion element disrupts the gene, ablates ApxIV expression, and prevents ApxIV-based serological detection (Tegetmeyer et al., 2008). Depending on the primers used, such insertions can affect the results of *apxIVA*-based PCR assays, possibly leading to misinterpretation (5, 17). The availability of multiple species-specific markers is therefore desirable to ensure accurate detection and prevent diagnostic evasion.

Previously, *A. pleuropneumoniae*-specific markers were discovered empirically by cross-species hybridization or PCR in which DNA fragments that can serve as species-specific markers were identified (13, 15, 16, 20). Now, with growing numbers of genome sequences of various pathogens available in public databases, these genome assemblies can be utilized for identification of new species-specific DNA markers for diagnostic purpose. Using genome sequence data to identify species-specific markers is superior to empirical testing of DNA fragments, since the content of whole genome can be screened comprehensively *in silico*, covering more putative markers and potentially yielding more species-specific markers. In this study, whole-genome sequences of *A. pleuropneumoniae* were mined for novel *A. pleuropneumoniae*-specific markers. The new markers identified can serve as alternative or additional markers in *A. pleuropneumoniae*-specific diagnostics.

**TABLE 1** Accession numbers of *A. pleuropneumoniae* complete genome assemblies used for identification of *A. pleuropneumoniae*-conserved sequences[a]

| No. | Strain | Serovar | Accession no. | Genome size (Mb) | Reference |
|---|---|---|---|---|---|
| 1 | ATCC 27088[T] | 1 | CP030753.1 | 2.32 | (47) |
| 2 | ATCC 27088[T] | 1 | CP029003.1 | 2.32 | (8) |
| 3 | KL16 | 1 | CP022715.1 | 2.37 | (48) |
| 4 | CCUG 47657 | 2 | LR134515.1 | 2.33 | |
| 5 | JL03 | 3 | CP000687.1 | 2.24 | (49) |
| 6 | ATCC 33378 | 4 | LS483358.1 | 2.34 | |
| 7 | L20 | 5b | CP000569.1 | 2.27 | (50) |
| 8 | App6 | 5 | CP026009.1 | 2.41 | |
| 9 | AP76 | 7 | CP001091.1 | 2.35 | |
| 10 | MIDG2331 | 8 | LN908249.1 | 2.34 | (51) |
| 11 | 405 | 8 | CP078508.1 | 2.32 | |

[a]ATCC, american type culture collection; CCUG, culture collection university of gothenburg; [T] indicates type strain of the species.

## RESULTS

**In silico identification of novel *A. pleuropneumoniae*-specific DNA marker candidates.** In order to identify new *A. pleuropneumoniae*-specific DNA markers, 11 complete *A. pleuropneumoniae* genome assemblies (Table 1) with sizes ranging between 2.24 – 2.41 Mb covering 7 serovars (serovars 1–5, 7, and 8) were analyzed. *A. pleuropneumoniae*-conserved sequences of 100 – 400 nucleotides sharing 100% identity among the 11 genomes were identified. Using MegaBLAST searches against the nucleotide collection (nr/nt) database and the WGS database of *Pasteurellaceae*, which include 34 *A. pleuropneumoniae* genomes covering 13 serovars, 30 genomes of 13 other *Actinobacillus* and [*Actinobacillus*] species, 116 genomes of *G. parasuis*, and 291 genomes of *P. multocida* (Table 2), 12 *A. pleuropneumoniae*-conserved sequences were shown to be specific to *A. pleuropneumoniae in silico* (Table 3). These 12 sequences are called "*A. pleuropneumoniae*-specific marker candidates." Each of the marker candidates are highly conserved among *A. pleuropneumoniae* genomes (Table 3). Five *A. pleuropneumoniae*-specific marker candidates are within the *apxIVA* gene, a known *A. pleuropneumoniae*-specific marker (9, 20), validating our *in silico* marker identification method as effective. In addition to the five *apxIVA* sequences, seven sequences also fit the criteria of being *A. pleuropneumoniae*-specific marker candidates. These seven sequences are within the *eamA*, *nusG*, *sppA*, *xerD*, *ybbN*, *ycfL*, and *ychJ* genes (Table 3). Nucleotide sequences of these markers are shown in supplemental material. The presence and specificity of these marker candidates were further validated by PCR.

**Molecular validation of novel *A. pleuropneumoniae*-specific markers.** As the *apxIVA* gene, whose sequence is unique to *A. pleuropneumoniae*, is a proven *A. pleuropneumoniae*-specific marker (7, 9, 10, 17), we did not perform PCR to validate the five marker candidates within the *apxIVA* gene. The presence of seven other marker candidates in *A. pleuropneumoniae* and other bacteria was examined by PCR using primers specific to each marker candidate and specific to *A. pleuropneumoniae* genomes *in silico* (Table 4). Genomic DNA from reference strains of *A. pleuropneumoniae* covering all 19 serovars, 108 *A. pleuropneumoniae* field isolates covering serovars 1, 2, 5, 12, 15, and nontypables, eight other *Actinobacillus* and [*Actinobacillus*] species, and seven other bacterial species was used as PCR template. For all seven marker candidates (i.e., those within the *eamA*, *nusG*, *sppA*, *xerD*, *ybbN*, *ycfL*, and *ychJ* genes), PCR amplicons of expected sizes were detected in all *A. pleuropneumoniae* strains and isolates but were absent in other species (Table 5). Representative gel electrophoresis results are also shown in supplemental material. As controls, two pairs of previously reported *apxIVA* primers (9) were tested and shown to be *A. pleuropneumoniae*-specific, as expected, recognizing all *A. pleuropneumoniae* strains and isolates tested (Table 5). These results indicate that the seven sequences within the *eamA*, *nusG*, *sppA*, *xerD*, *ybbN*, *ycfL*, and *ychJ* genes are validated as novel *A. pleuropneumoniae*-specific DNA markers, can serve as additional or

**TABLE 2** Number of genome assemblies of selected species from the *Pasteurellaceae* family or of other pig pathogens in the NCBI databases available for *in silico* comparison[a]

| Species | No. of total genome assemblies (in the nr/nt and WGS databases) | No. of complete genome assemblies (in the nr/nt database) |
|---|---|---|
| *Actinobacillus capsulatus* | 1 | 0 |
| [*Actinobacillus*] *delphinicola* | 1 | 1 |
| *Actinobacillus equuli* | 3 | 3 |
| [*Actinobacillus*] *indolicus* | 3 | 1 |
| *Actinobacillus lignieresii* | 3 | 1 |
| [*Actinobacillus*] *minor* | 2 | 0 |
| *Actinobacillus pleuropneumoniae* | 34 | 11 |
| [*Actinobacillus*] *porcinus* | 2 | 0 |
| [*Actinobacillus*] *porcitonsillarum* | 1 | 1 |
| [*Actinobacillus*] *seminis* | 2 | 0 |
| [*Actinobacillus*] *succinogenes* | 1 | 1 |
| *Actinobacillus suis* | 7 | 2 |
| *Actinobacillus ureae* | 3 | 0 |
| *Actinobacillus vicugnae* | 1 | 0 |
| *Aggregatibacter actinomycetemcomitans* | 97 | 12 |
| *Bibersteinia trehalosi* | 7 | 4 |
| *Escherichia coli* | 24529 | 1782 |
| *Glaesserella parasuis* | 116 | 24 |
| *Haemophilus haemolyticus* | 68 | 4 |
| *Haemophilus influenzae* | 779 | 73 |
| *Haemophilus parainfluenzae* | 99 | 16 |
| Influenza A virus | 130 | 127 |
| *Mannheimia haemolytica* | 196 | 85 |
| *Pasteurella multocida* | 291 | 81 |
| *Salmonella enterica* | 12336 | 1066 |
| *Streptococcus suis* | 1623 | 72 |

[a]*E. coli*, Influenza A virus, *S. enterica*, and *S. suis* are not in the *Pasteurellaceae* family; therefore, only their complete genomes in the nr/nt database were used for *in silico* marker specificity test. Species with [*Actinobacillus*] are not officially included in the *Actinobacillus* genus, but have not yet been assigned to a new genus (25).

alternative targets for *A. pleuropneumoniae* detection assays, and are interchangeable with *apxIVA*. The use of more than one marker can prevent diagnostic evasion.

## DISCUSSION

Identifying species-specific markers for *A. pleuropneumoniae* previously involved individually testing DNA fragments in cross-hybridization or PCR experiments (9, 13, 15, 20). These methods are time-consuming and incomprehensive, as only a limited number of DNA fragments can be tested. Moreover, some of the resulting detection assays are not species-specific and still show cross-reactivity with closely related species (3, 13, 15). Using comparative genome analysis, based on a strict criterion of 100% nucleotide identity across sequences of 100–400 nucleotides conserved in only 11 complete *A. pleuropneumoniae* genomes, 12 sequences were identified as putatively *A. pleuropneumoniae*-specific (Table 3). Other highly conserved sequences with less than 100% conservation among the 11 complete genomes were not considered here but may be useful as *A. pleuropneumoniae*-specific markers and require further investigation.

Even though 11 complete *A. pleuropneumoniae* genome assemblies covering serovars 1–5 and 7–8 (Table 1) were used for the initial step of *A. pleuropneumoniae*-conserved sequences identification, the *A. pleuropneumoniae*-conserved sequences were later tested for their *A. pleuropneumoniae*-specificity using the nr/nt nucleotide collection database and the WGS database limited to the *Pasteurellaceae* family, which include 34 complete and incomplete *A. pleuropneumoniae* genome assemblies covering serovars 1–13, 30 genome assemblies from 13 other *Actinobacillus* and [*Actinobacillus*] species, 116 *G. parasuis* genome assemblies, and 291 *P. multocida* genome assemblies (Table 2). Since large

**TABLE 3** *A. pleuropneumoniae*-specific DNA marker candidates identified *in silico*[a]

| No. | Target | Locus tag in L20 (CP000569) | Predicted function | Length (NTs) | Match to 11 complete *A. pleuropneumoniae* genomes | | Match to incomplete *A. pleuropneumoniae* genomes | | |
|---|---|---|---|---|---|---|---|---|---|
| | | | | | % query cover | % identity | No. of matches in 23 incomplete *A. pleuropneumoniae* genomes | % query cover | % identity |
| 1 | *apxIVA*-1 | APL_0998 | Toxin | 385 | 100 | 100 | 51 (match more than 1 contig in a genome) | 19-100 | 79.43-100 |
| 2 | *apxIVA*-2 | APL_0998 | Toxin | 125 | 100 | 100 | 19 | 38-100 | 96.8-100 |
| 3 | *apxIVA*-3 | APL_0998 | Toxin | 326 | 100 | 100 | 23 | 96-100 | 99.08-100 |
| 4 | *apxIVA*-4 | APL_0998 | Toxin | 315 | 100 | 100 | 23 | 100 | 100 |
| 5 | *apxIVA*-5 | APL_0998 | Toxin | 116 | 100 | 100 | 23 | 100 | 100 |
| 6 | *eamA* | APL_1023 | EamA family transporter; DMT family transporter | 203 | 100 | 100 | 23 | 100 | 99.51−100 |
| 7 | *nusG* | APL_1717 | Transcription termination/anti-termination protein | 139 | 100 | 100 | 23 | 100 | 100 |
| 8 | *sppA* | APL_1268 | Signal peptide peptidase, protease IV | 105 | 100 | 100 | 23 | 100 | 100 |
| 9 | *xerD* | APL_1542 | Site-specific tyrosine recombinase | 149 | 100 | 100 | 22 (absent in contigs of ATCC 33377) | 100 | 100 |
| 10 | *ybbN* | APL_0080 | Cochaperone YbbN; putative thioredoxin-like protein | 127 | 100 | 100 | 23 | 100 | 100 |
| 11 | *ycfL* | APL_0125 | YcfL family protein; putative periplasmic lipoprotein | 101 | 100 | 100 | 23 | 100 | 100 |
| 12 | *ychJ* | APL_1658 | YchJ family protein, hypothetical protein, SEC-C motif containing | 140 | 100 | 100 | 24 (present twice in strain 4226) | 100 | 99.29−100 |

[a]Percent query cover and percent identity after performing MegaBLAST searches against the nr/nt or whole-genome sequence (WGS) databases are shown. No similarity between marker candidates and sequences from other species was found by MegaBLAST.

**TABLE 4** Primers used in this study

| Primer no. | Primer name | Sequence (5′ to 3′) |
|---|---|---|
| P228 | *eamA*-F | CACTTCAAGTCGGCACTGTC |
| P229 | *eamA*-R | TCATAATAATTGCAGCGTTAGTGA |
| P230 | *sppA*-F | CCAACGACGTAAAGCGAATAA |
| P231 | *sppA*-R | CGAACAGACTATCGTCGCT |
| P240 | *xerD*-F | ATAACGTATCTAAAAACTGTTCG |
| P241 | *xerD*-R | TAGAATATCTAGGAATAAAAGTAGC |
| P242 | *ychJ*-F | CGGTTATTTTTTCAAAATTCTTTGC |
| P243 | *ychJ*-R | CGCCTATTTAGCCTAATCC |
| P250 | *nusG*-F | GGCTTTGTGATTTTATAAAATAAG |
| P251 | *nusG*-R | GCCGATAAAAAACACTTTGTG |
| P254 | *ybbN* -F | TCATTATTACGCCGGTTGGC |
| P255 | *ybbN* -R | TCACGGTTGCCAATAAAAATTG |
| P256 | *ycfL*-F | ACTCAACCAAGGTTGCATCG |
| P257 | *ycfL*-R | AATCAAGGCATTACACAAACCAA |
|  | ApxIVA-1L | TGGCACTGACGGTGATGA (9) |
|  | ApxIVA-1R | GGCCATCGACTCAACCAT (9) |
|  | ApxIVANEST-1L | GGGGACGTAACTCGGTGATT (9) |
|  | ApxIVANEST-1R | GCTCACCAACGTTTGCTCAT (9) |

genome databases can be accessed and utilized, *in silico* genome analysis is a powerful tool to guide marker discovery. The more genomes of target species and closely related nontarget species become available, the higher accuracy and specificity of *in silico* marker discovery will be. Molecular validation is still necessary, especially for marker discovery of species with limited genome data. The more bacterial species and isolates that are available for molecular validation, the more accurate and specific the resulting markers will be.

Five marker candidates identified in this study are within *apxIVA*, previously reported to be an *A. pleuropneumoniae*-specific gene (9, 20), confirming that our *in silico* marker identification method is effective. Nonetheless, the five *apxIVA* sequences (*apxIVA*-1-5) identified in this study are not identical to those previously described. As there have been reports of atypical *A. pleuropneumoniae* isolates failing to amplify the predicted target with existing *apxIVA*-specific primers (5, 21), our new *apxIVA* targets provide alternative options for molecular confirmation of *A. pleuropneumoniae*.

Two *apxIVA* regions (*apxIVA*-1 and 2) identified herein are in the 3′ part of *apxIVA* and are in close proximity to the *A. pleuropneumoniae*-specific region previously identified in hybridization experiments and some previously published primer pairs (Fig. 1A) (7, 9, 20). The 5′ and central parts of the *apxIVA* gene were originally disregarded as *A. pleuropneumoniae*-specific because probes from these regions showed weak hybridization signals with *A. lignieresii* (9, 20). However, later studies identified additional conserved regions within the 5′ (17) and the central part of *apxIVA* (10) that can be used as targets for *A. pleuropneumoniae* molecular detection assays (Fig. 1A). Three newly identified marker candidates (*apxIVA*-3, 4, and 5) are within the central part of *apxIVA* (Fig. 1), but do not overlap the conserved regions previously reported (10, 17), as these sequences do not match our criteria of being 100% conserved among the 11 complete genomes. These combined results indicate that our marker discovery strategy does not identify all possible markers but is useful for identifying multiple effective species-specific markers simultaneously.

In addition to *apxIVA*, some *A. pleuropneumoniae* strains also contain *apxIV*-S, a partial duplication of *apxIVA* that shares approximately 87% identity with *apxIVA* in the 3′ region (Fig. 1B) (22). In *A. pleuropneumoniae* genomes with both *apxIVA* and *apxIV*-S, the five new *apxIVA* marker candidates match to different regions but are still *A. pleuropneumoniae*-specific *in silico* (Fig. 1B, Table 3). Regions *apxIVA*-1′ and 2′ with 90% and 94–98% identity to *apxIVA*-1 and *apxIVA*-2, respectively, are also present (Fig. 1). Coamplification of *apxIVA*-1′ and *apxIVA*-2′ along with *apxIVA*-1 and *apxIVA*-2 is

**TABLE 5** Validation of *A. pleuropneumoniae*-specific markers by PCR[a]

| Species | Serovar | Strain | No. of strains tested | apxIVA | | Marker candidate | | | | | | |
|---|---|---|---|---|---|---|---|---|---|---|---|---|
| | | | | 1L-1R (422) | NEST 1L-1R (377) | eamA (192) | nusG (117) | sppA (83) | xerD (74) | ybbN (58) | ycfL (54) | ychJ (66) |
| A. pleuropneumoniae | 1 | ATCC 27088ᵀ, 2 field isolates | 3 | ++ | ++ | ++ | ++ | ++ | ++ | ++ | ++ | ++ |
| | 2 | ATCC 27089, 1 field isolate | 2 | ++ | ++ | ++ | ++ | ++ | ++ | ++ | ++ | ++ |
| | 3 | ATCC 27090 | 1 | ++ | ++ | ++ | ++ | ++ | ++ | ++ | ++ | ++ |
| | 4 | ATCC 33378 | 1 | ++ | ++ | ++ | ++ | ++ | ++ | ++ | ++ | ++ |
| | 5 | ATCC 33377, L20, ATCC 55454, 100 field isolates | 103 | ++ | ++ | ++ | ++ | ++ | ++ | ++ | ++ | ++ |
| | 6 | ATCC 33590 | 1 | ++ | ++ | ++ | ++ | ++ | ++ | ++ | ++ | ++ |
| | 7 | WF83 | 1 | ++ | ++ | ++ | ++ | ++ | ++ | ++ | ++ | ++ |
| | 8 | 405 | 1 | ++ | ++ | ++ | ++ | ++ | ++ | ++ | ++ | ++ |
| | 9 | CVJ13261 | 1 | ++ | ++ | ++ | ++ | ++ | ++ | ++ | ++ | ++ |
| | 10 | D13039 | 1 | ++ | ++ | ++ | ++ | ++ | ++ | ++ | ++ | ++ |
| | 11 | 56153 | 1 | ++ | ++ | ++ | ++ | ++ | ++ | ++ | ++ | ++ |
| | 12 | 8328, 1 field isolate | 2 | ++ | ++ | ++ | ++ | ++ | ++ | ++ | ++ | ++ |
| | 13 | N-273 | 1 | ++ | ++ | ++ | ++ | ++ | ++ | ++ | ++ | ++ |
| | 14 | 3906 | 1 | ++ | ++ | ++ | ++ | ++ | ++ | ++ | ++ | ++ |
| | 15 | HS143, 1 field isolate | 2 | ++ | ++ | ++ | ++ | ++ | ++ | ++ | ++ | ++ |
| | 16 | A-85/14 | 1 | ++ | ++ | ++ | ++ | ++ | ++ | ++ | ++ | ++ |
| | 17 | 16287-1 | 1 | ++ | ++ | ++ | ++ | ++ | ++ | ++ | ++ | ++ |
| | 18 | 7311555 | 1 | ++ | ++ | ++ | ++ | ++ | ++ | ++ | ++ | ++ |
| | 19 | 7213384-1 | 1 | ++ | ++ | ++ | ++ | ++ | ++ | ++ | ++ | ++ |
| | Nontypable | 3 field isolates | 3 | ++ | ++ | ++ | ++ | ++ | ++ | ++ | ++ | ++ |
| A. equuli | | ATCC 9346 | 1 | - | - | - | - | - | - | - | - | - |
| [A.] indolicus | | CCUG 39029ᵀ | 1 | - | - | - | - | - | - | - | - | - |
| A. lignieresii | | ATCC 13372, CCUG 41384ᵀ | 2 | - | - | - | - | - | - | - | - | - |
| [A.] minor | | CCUG 38923ᵀ | 1 | - | - | - | - | - | - | - | - | - |
| [A.] porcinus | | CCUG 38924ᵀ | 1 | - | - | - | - | - | - | - | - | - |
| [A.] rossi | | ATCC 27072 | 1 | - | - | - | - | - | - | - | - | - |
| A. suis | | ATCC 15557, ATCC 33415ᵀ | 2 | - | - | - | - | - | - | - | - | - |
| A. ureae | | ATCC 25976 | 1 | - | - | - | - | - | - | - | - | - |
| B. trehalosi | | ATCC 33367 | 1 | - | - | - | - | - | - | - | - | - |
| G. parasuis | | ATCC 19417 | 1 | - | - | - | - | - | - | - | - | - |
| | | Field isolates | 6 | - | - | - | - | - | - | - | - | - |
| H. influenzae | | ATCC 33391 | 1 | - | - | - | - | - | - | - | - | - |
| M. haemolytica | | ATCC 29696 | 1 | - | - | - | - | - | - | - | - | - |
| P. multocida | | ATCC 43137 | 1 | - | - | - | - | - | - | - | - | - |
| | | ATCC BAA-1113 | 1 | - | - | - | - | - | - | - | - | - |
| S. Choleraesuis | | ATCC 7001 | 1 | - | - | - | - | - | - | - | - | - |
| S. suis | | ATCC 43765 | 1 | - | - | - | - | - | - | - | - | - |

[a]++, PCR product of expected size was present; -, no PCR product present; numbers in parentheses are expected PCR product sizes in base pairs (bp). Genomic DNA of various bacterial species/strains was tested for the presence of candidate marker sequences using PCR.

possible but does not alter PCR product sizes and thus detection results. The presence of multiple highly homologous regions in one genomic DNA molecule may serve as more targets for PCR, possibly leading to detection assays with higher sensitivity.

Although not encoding an intact ApxIV protein (NCBI accession no. NZ_LR134169), the NCTC 10568 *A. lignieresii* genome contains sequences (comprising multiple open reading frames) sharing 73% identity over 71% of the *A. pleuropneumoniae apxIVA* sequence (71% query cover), as determined by BLASTn. Five *A. pleuropneumoniae*-specific *apxIVA* marker candidates identified here do not share significant similarity with the *apxIVA*-like sequences in *A. lignieresii*, as determined by default parameters of

## A. *apxIVA* in CCUG 47657

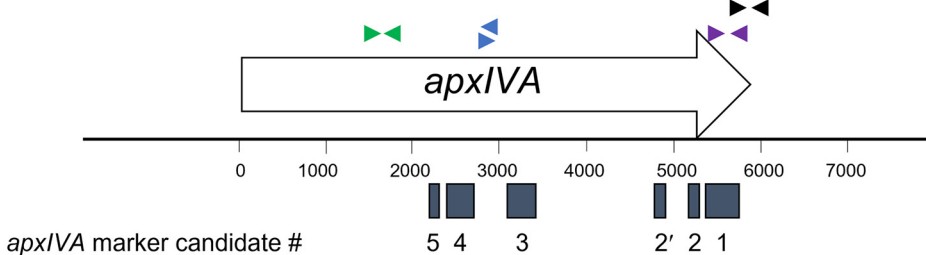

## B. *apxIVA* and *apxIV-S* in MIDG2331

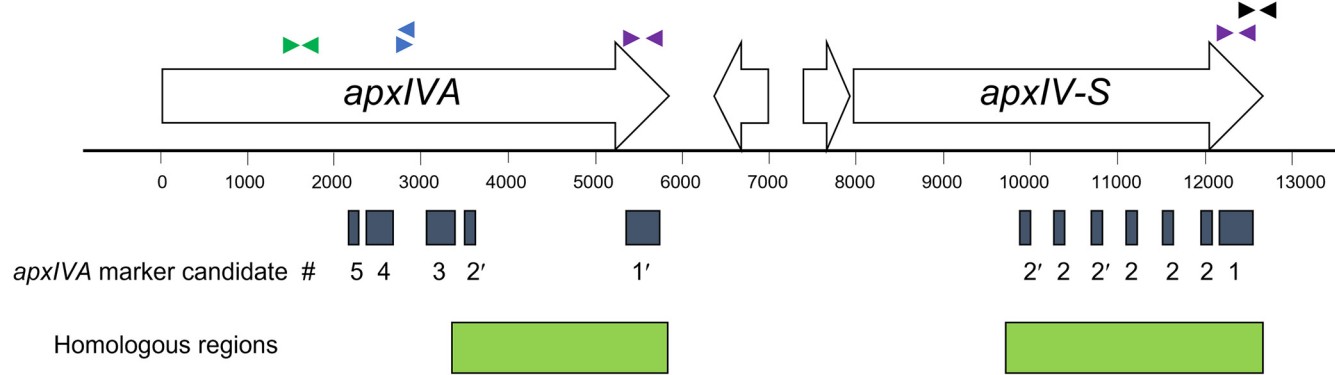

**FIG 1** Locations of previously published primer pairs and newly identified *apxIVA* marker candidates in the genome of the *A. pleuropneumoniae* serovar 2 strain CCUG 47657 that contains only the *apxIVA* gene (A), and in the genome of the *A. pleuropneumoniae* serovar 8 strain MIDG2331 that contains both *apxIVA* and *apxIV-S* genes (B). Previously published primer pairs are shown as arrowheads. Green arrowheads denote primers oAPXIVA-TSP1 and oAPXIVA-TSP2 (17). Blue arrowheads denote primers apxIVA-exo-F and apxIVA-exo-R (10). Purple arrowheads denote primers ApxIVA-1L and ApxIVA-1R (9). Black arrowheads denote primers named apxIVA1 and apxIVA3 (7). Gray rectangles represent regions *apxIVA*-1 to *apxIVA*-5 identified in this study (Table 3). Region *apxIVA*-1' is 90% identical to *apxIVA*-1. Region *apxIVA*-2' is 94 to 98% identical to region 2. Green rectangles indicate homologous regions between *apxIVA* and *apxIV-S*.

MegaBLAST search against databases which include three complete and incomplete *A. lignieresii* genomes (Table 3). In short, five *A. pleuropneumoniae*-specific *apxIVA* regions are *A. pleuropneumoniae*-specific despite the presence of *apxIVA*-like sequences in *A. lignieresii*. Nonetheless, cross-reactivity with *A. lignieresii* in pig-derived samples is unlikely, as *A. lignieresii* is a pathogen of cattle and sheep (23).

In addition to sequences within *apxIVA*, seven novel marker candidates that map to various genes were identified. Six newly identified *A. pleuropneumoniae*-specific markers, namely, *eamA*, *nusG*, *sppA*, *ybbN*, *ycfL*, and *ychJ*, share 100% identity among all 11 complete *A. pleuropneumoniae* genome assemblies and 99.29–100% identity among all 23 incomplete *A. pleuropneumoniae* genome assemblies, confirming their highly conserved nature among *A. pleuropneumoniae* genomes. These six sequences are also *A. pleuropneumoniae*-specific compared *in silico* with available databases (Table 3) and when tested by PCR with DNA from available bacterial species and strains (Table 5).

The last marker candidate, *xerD*, shares 100% identity among all 11 complete *A. pleuropneumoniae* genomes but is found only in 22 out of 23 incomplete *A. pleuropneumoniae* genomes. The *xerD* marker candidate is absent in genome contigs of the ATCC 33377 strain (CABEFA01), suggesting that the ATCC 33377 genome may not contain *xerD* or the contigs that contain whole *xerD* marker sequence are absent in the genome assemblies. The *xerD* sequence identified is only 149 nucleotides in length. Assembling contigs to contain this short sequence should not be difficult unless the genome contains multiple sequences homologous to *xerD*. As seen in the case of *apxIVA*-2, when performing MegaBLAST searches against the *Pasteurellaceae* WGS database, only 19 out of 23 matches with *A. pleuropneumoniae* incomplete genomes (38–100% query cover and 96.8–100% identity) were observed (Table 3). Nonetheless, *xerD*-specific PCR product

**TABLE 6** Bacteria used in this study[a]

| Genus and species | Serovar | Strain name | Source/reference[c] |
|---|---|---|---|
| *Actinobacillus pleuropneumoniae* | 1 | ATCC 27088[T] | ATCC (33) |
| | 2 | ATCC 27089 | ATCC (33) |
| | 3 | ATCC 27090 | ATCC (33) |
| | 4 | ATCC 33378 | ATCC (34) |
| | 5a | ATCC 33377 | ATCC (34, 35) |
| | 5b | L20 | (34, 35) |
| | 5 | ATCC 55454 | ATCC |
| | 6 | ATCC 33590 | ATCC (36) |
| | 7 | WF83 | (37) |
| | 8 | 405 | (38) |
| | 9 | CVJ13261 | (39) |
| | 10 | D13039 | (40) |
| | 11 | 56153 | (41) |
| | 12 | 8328 | Denmark |
| | 13 | N-273 | (42) |
| | 14 | 3906 | (43) |
| | 15 | HS143 | (44) |
| | 16 | A-85/14 | (45) |
| | 17 | 16287-1 | (46) |
| | 18 | 7311555 | (46) |
| | 19 | 7213384-1 | (5) |
| | 1 [2][b] | Field isolates from Thailand [108] | This study |
| | 2 [1] | | |
| | 5 [100] | | |
| | 12 [1] | | |
| | 15 [1] | | |
| | Nontypable [3] | | |
| *Actinobacillus equuli* | | ATCC 9346 | ATCC |
| [*Actinobacillus*] *indolicus* | | CCUG 39029[T] | CCUG |
| *Actinobacillus lignieresii* | | ATCC 13372 | ATCC |
| | | CCUG 41384[T] | CCUG |
| [*Actinobacillus*] *minor* | | CCUG 38923[T] | CCUG |
| [*Actinobacillus*] *porcinus* | | CCUG 38924[T] | CCUG |
| [*Actinobacillus*] *rossi* | | ATCC 27072 | ATCC |
| *Actinobacillus suis* | | ATCC 15557 | ATCC |
| | | ATCC 33415[T] | ATCC |
| *Actinobacillus ureae* | | ATCC 25976 | ATCC |
| *Bibersteinia trehalosi* | | ATCC 33367 | ATCC |
| *Glaesserella parasuis* | | ATCC 19417 | ATCC |
| | | Field isolates from Thailand [6] | This study |
| *Haemophilus influenzae* | | ATCC 33391 | ATCC |
| *Mannheimia haemolytica* | | ATCC 29696 | ATCC |
| *Pasteurella multocida* | | ATCC 43137 | ATCC |
| | | ATCC BAA-1113 | ATCC |
| *Salmonella enterica* subsp. *enterica* | Choleraesuis | ATCC 7001 | ATCC |
| *Streptococcus suis* | | ATCC 43765 | ATCC |

[a]ATCC, american type culture collection; CCUG, culture collection university of gothenburg.
[b]Numbers in brackets indicate the number of isolates. [T] indicates type strain of the species. Species with [*Actinobacillus*] are not officially included in the *Actinobacillus genus*, but have not yet been assigned to a new genus (25).
[c]The Langford laboratory was the source of bacteria (or gDNA) that were not purchased from ATCC or CCUG. The growth and preparation of derived gDNA from these strains was carried out as described previously (5).

was observed when genomic DNA from the ATCC 33377 strain was used as the template (Table 5 and 6), indicating that *xerD* can also serve as a marker for *A. pleuropneumoniae* identification.

The use of multiple targets in a diagnostic assay can reduce false-negative results among *A. pleuropneumoniae* strains that may evade current detection methods. These

novel *A. pleuropneumoniae*-specific markers could serve as targets for other DNA amplification assays such as isothermal amplification assays, which are more field-ready than PCR.

In conclusion, this study demonstrates how comparative genomics and molecular validation can accelerate species-specific marker discovery, save time, labor, and cost, and result in more markers compared with traditional marker discovery by hybridization or PCR experiments. The marker discovery strategy described herein can be applied to other species with sufficient genome data, leading to novel markers and diagnostic assays for infectious diseases.

## MATERIALS AND METHODS

The experiments using *Actinobacillus* and other bacterial species were approved by BIOTEC and Chulalongkorn University Institutional Review Boards on Biosafety and Biosecurity with approval numbers BT-IBC-61-026 and IBC1831058, respectively.

**A. pleuropneumoniae isolation from clinical samples.** *A. pleuropneumoniae* was isolated from lung or pleural fluid samples from pigs with clinical signs of respiratory disease submitted to the Livestock Animal Hospital, Chulalongkorn University, Nakhon Pathom, Thailand during 2017–2018, as per standard techniques (24). Briefly, clinical samples were cultured on blood agar (containing 5% sheep red blood cells) with a *Staphylococcus aureus* nurse streak and incubated at 37°C with 5% $CO_2$. Hemolytic colonies with a satellite characteristic around the *S. aureus* streak were further tested by Gram staining, Christie–Atkins–Munch-Peterson (CAMP) reaction with *S. aureus*, and catalase and oxidase tests. Species validation and molecular serotyping were performed using multiplex PCR targeting *apxIVA* and *cps* genes (7).

**Bacterial strains and growth conditions.** Bacterial strains used to test the presence of DNA markers in this study are either in the *Pasteurellaceae* family or are present in pigs as commensal or pathogenic bacteria (Table 6). Bacteria (or genomic DNA) were purchased from the American Type Culture Collection (ATCC) or Culture Collection of University of Gothenburg (CCUG) or obtained from the Langford laboratory as indicated (Table 6). Some [*Actinobacillus*] species such as [*A.*] indolicus, [*A.*] minor, and [*A.*] porcinus are not officially included in the *Actinobacillus* genus but have not yet been assigned to a new genus (25). These species are herein described as [*Actinobacillus*.] *Actinobacillus* and [*Actinobacillus*] species, *Glaesserella parasuis*, *Pasteurella multocida*, and *Haemophilus influenzae* were grown on chocolate blood agar supplemented with IsoVitalex (BBL, BD, Franklin Lakes, NJ, USA) at 37°C with 5% $CO_2$. *Bibersteinia trehalosi*, *Mannheimia haemolytica*, *Salmonella enterica* serovar Choleraesuis, and *Streptococcus suis* were grown on brain heart infusion (BHI) plates at 37°C with 5% $CO_2$.

**In silico DNA marker identification.** In the initial step, 11 complete genome assemblies covering serovars 1–5 and 7–8 (Table 1) were selected for analysis in consideration for algorithm efficiency. Sequences of 100 – 400 nucleotides in length that share 100% identity among the 11 complete genomes were selected by a custom script as "*A. pleuropneumoniae*-conserved sequences." In the second step, these *A. pleuropneumoniae*-conserved sequences were used as queries to search for highly similar sequences using MegaBLAST (26–28). Searches were performed against the nucleotide collection (nr/nt) database, which contains sequences from GenBank, EMBL, DDBJ, PDB, and RefSeq, but excludes draft whole-genome contigs (WGS). Nineteen *A. pleuropneumoniae*-conserved sequences were identified to be specific to *A. pleuropneumoniae* genomes compared with the nr/nt database. In the third step, these 19 *A. pleuropneumoniae*-conserved sequences were used as queries to search for highly similar sequences in the WGS database containing draft genome contigs, limited to sequences of the *Pasteurellaceae* family, using MegaBLAST. Twelve *A. pleuropneumoniae*-conserved sequences remained specific to *A. pleuropneumoniae* genomes *in silico* compared with the WGS database and were considered "*A. pleuropneumoniae*-specific marker candidates." The number of genome assemblies of selected species (in the same family as *A. pleuropneumoniae* or also present in pigs) available for *in silico* comparison is shown in Table 2.

**In silico primer design.** BLASTn (26, 27), suitable for identification of more dissimilar sequences, was used to identify sequences of non-*A. pleuropneumoniae* species that share more than 70% identity with *A. pleuropneumoniae*-specific marker candidates from the nucleotide collection (nr/nt) database. Multiple alignment of *A. pleuropneumoniae*-specific marker candidates and similar sequences from other species was performed using Clustal Omega (29, 30). Regions with high mismatch between *A. pleuropneumoniae* and non-*A. pleuropneumoniae* species were selected for PCR primer design. Primer BLAST (31) searches against the nr/nt database were used to confirm that the newly designed PCR primers (Table 4) yielded PCR products of expected size only when *A. pleuropneumoniae* genomes were used as template.

**Genomic DNA purification and PCR amplification.** Genomic DNA of various bacterial species was extracted using a standard DNA purification protocol (32). PCR was performed using *Taq* DNA polymerase with Standard *Taq* Buffer (M0273, New England Biolabs, Ipswich, MA, USA) according to the manufacturer's protocol. Briefly, PCRs were prepared to contain final concentrations of 200 $\mu$M dNTPs, 0.2 $\mu$M each primer (Table 4), 0.025 U/$\mu$l *Taq* DNA polymerase, and 1 ng/$\mu$L of bacterial genomic DNA. Thirty cycles of 95°C for 30 s, 60°C for 1 min, and 68°C for 1 min were performed using a C1000 Touch PCR Thermal Cycler (Bio-Rad, Hercules, CA, USA). PCR products were visualized by agarose gel electrophoresis followed by ethidium bromide staining. Alternatively, Luna qPCR Master Mix (M3003, New England Biolabs) was used according to the manufacturer's protocol. Briefly, qPCRs were prepared to contain final concentrations of 0.25 $\mu$M each primer and 1 ng/$\mu$L of bacterial genomic DNA. Forty-five cycles of

95°C for 15 s and 60°C for 30 s were performed using a Bio-Rad CFX96 real-time PCR machine. Fluorescence signals indicative of the presence of PCR products were measured.

## SUPPLEMENTAL MATERIAL

Supplemental material is available online only.

**SUPPLEMENTAL FILE 1**, PDF file, 0.3 MB.

## ACKNOWLEDGMENTS

This study was funded by grants from National Center for Genetic Engineering and Biotechnology (BIOTEC) (P18-50442), National Science and Technology Development Agency (NSTDA) (P20-50968), Thammasat University Research Fund under the TU Research Scholar, Contract No. TP 2/24/2560, and the UK BBSRC (BB/S002103/1). U.L. was supported by BIOTEC (P16-52034) and NSTDA (P20-50077). We thank Philip J. Shaw for suggestions on the manuscript.

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
