## [Reviewer comments · Microbiology Spectrum]

Microbiology Spectrum

Novel DNA markers for identification of *Actinobacillus pleuropneumoniae*

Gun Srijuntongsiri, Atiwat Mhoowai, Sukuma Samngamnim, Pornchalit Assavacheep, Janine Bossé, Paul Langford, Navaporn Posayapisit, Ubolsree Leartsakulpanich, and Warangkhana Songsungthong

Corresponding Author(s): Warangkhana Songsungthong, National Center for Genetic Engineering and Biotechnology

Review Timeline:

Submission Date:	August 23, 2021
Editorial Decision:	October 20, 2021
Revision Received:	November 13, 2021
Accepted:	December 2, 2021

Editor: Sandeep Tamber

Reviewer(s): Disclosure of reviewer identity is with reference to reviewer comments included in decision letter(s). The following individuals involved in review of your submission have agreed to reveal their identity: Jinlin Liu (Reviewer #2)

Transaction Report:

DOI: <https://doi.org/10.1128/Spectrum.01311-21>

October 20, 2021

Dr. Warangkhanha Songsungthong
National Center for Genetic Engineering and Biotechnology
113 Thailand Science Park
Klong Luang, Pathum Thani 12120
Thailand

Re: Spectrum01311-21 (Novel DNA markers for detecting *Actinobacillus pleuropneumoniae*)

Dear Dr. Warangkhanha Songsungthong:

Thank you for submitting your manuscript to Microbiology Spectrum. When submitting the revised version of your paper, please provide (1) point-by-point responses to the issues raised by the reviewers as file type "Response to Reviewers," not in your cover letter, and (2) a PDF file that indicates the changes from the original submission (by highlighting or underlining the changes) as file type "Marked Up Manuscript - For Review Only". Please use this link to submit your revised manuscript - we strongly recommend that you submit your paper within the next 60 days or reach out to me. Detailed information on submitting your revised paper are below.

Link Not Available

Sincerely,

Sandeep Tamber

Journals Department
Reviewer comments:

Reviewer #1 (Comments for the Author):

This MS reports on the use of whole genome data mining to screen for novel species-specific DNA markers for a major pig pathogen - *Actinobacillus pleuropneumoniae*. While there is an existing range of *Actinobacillus pleuropneumoniae* DNA-based assays, this MS has the novelty of highlighting the emerging technology of whole genome data mining to develop novel assays.

I have the following comments

Major Comments

1. Title. In my view, the title is somewhat (albeit unintentionally) misleading. The work described in this MS is all about the use of molecular assays on pure isolated cultures ie identification. The use of these targets in the manner validated in this MS is actually not about "detection" as this word (at least in diagnostic laboratory settings) is about detecting directly in the infected animal. In my view, the title should use the word "identification" in place of "detection".
2. Taxonomy. The authors appear not to be aware of the now widely accepted situation that many of the bacterial species

identified in this MS as being in the genus *Actinobacillus* are clearly NOT in this genus. This continued allocation of these problematic organisms to the genus *Actinobacillus* when they are clearly not in the genus is only prolonging and intensifying the confusion. It would help all workers if those organisms that are clearly not members of the genus *Actinobacillus* be named at each and occasion with a clear flag that they are not members of the genus e.g. by placing the in-correct genus name in quotation marks ("*Actinobacillus*") or in square brackets ([*Actinobacillus*]). Based on the latest edition of Bergey's Manual (see Blackall, P.J., Turni, C. 2020. *Actinobacillus*. In Bergey's Manual of Systematics of Archaea and Bacteria, Whitman, W.B., Rainey, F., Kämpfer, P., Trujillo, M., Chun, J., DeVos, P., Hedlund, B., Dedysh, S., eds. (John Wiley & Sons), the following [*Actinobacillus*] species featured in this MS are clearly NOT members of the genus *Actinobacillus* and should always have the genus name clearly marked as either "*Actinobacillus*" or [*Actinobacillus*] with a similar approach for the abbreviated genus name "A." or [A.]:-

- a. "A." *delphinocola*
- b. "A." *indolicus*
- c. "A." *minor*
- d. "A." *porcinus*
- e. "A." *rossi*
- f. "A." *seminis*
- g. "A." *succinogenes*

For the moment, the final resolution of the position of "A." *porcitosillarum* also remains unclear and that species should also be flagged as well.

3. Lack of technical details. At lines 164-172, the text describes the PCRs used in this work. I found the text to be far too light on detail. There are many Taq enzymes supplied by NEB - which one was used? What were the cycling conditions of the assays?

4. Field isolates. Based on Table 1, this study included 108 field isolates. The text could be more informative by indicating the number of isolates per serovar (simply place the number in brackets after the serovar and explain via a footnote).

5. Table 1. This Table needs considerable refinement. The column headed "No." is irrelevant and the column should be deleted. The column headed Species is actually Genus and Species. The column headed Source/Reference is mostly presenting strain name and reference (ie ATCC 27088 in the first cell in this column is the strain name - the Source is simply ATCC. As well, the columns headed "Strain Name" and "Source/Reference" are largely duplicating each other in terms of the strain name. The column headed "Strain name" should record the ATCC or CCUG strain name (readers can consult the ATCC and CCUG web sites for the alternative names of each strain) for all strains obtained from CCUG and ATCC. The Source/Reference column should just be showing ATCC or CCUG (and not the strain name). What was the source of the bacteria without a source name e.g. *A. pleuropneumoniae* serovar 7? The column headed "Number of strains/isolate" is wasting valuable space. Delete this column and simply use a footnote to record the rare exception where more than a single strain/isolate was used. Where the formal type strain of species has been used the T signifying that the strain is the formal species type strain should be in superscript. ATCC 27088^T (where {} shows superscript) is the type strain for *A. pleuropneumoniae*.

6. Confusion on strain names. As I indicated above, there are problems with confusion over strain names in Table 1. This lack of completed understanding has spoiled over into the general text. At lines 280 - 293, there is Discussion about the absence of *xerD* in a strain called NCTC 11383 (note the space - should be used with all NCTC and ATCC strains). The authors appear not to understand that NCTC 11383 is actually K17. Based on Tables 1 and 6, the pure culture of NCTC 11383/ATCC 33377/K17 was used in this study as a live culture and was tested by PCR and was positive for *xerD*.

7. Confusion over strains. Given the confusion highlighted above, the authors need to go back into their data and look for other duplications of strains. There may well be additional duplications of strains that are actually the same strain but simply obtained from ATCC or NCTC or other sources.

8. Table 3. Again, remove the first column - it is irrelevant.

9. Table 5. Again, a poorly laid out Table. The column headed "nucleotide sequence" is simply too bulky and makes this table very hard for the reader to assimilate. Move the details in this column to a Supplementary Table. The column headed "Among WGS of Pasteurellaceae" is very confusing. At a guess, the data being shown in the three sub-columns here is actually limited to the similarities in the available incomplete *A. pleuropneumoniae* genomes. The last column is simply a waste of space and can be replaced by a footnote that records whatever was meant to be shown in this column (it is just empty cells at the moment). I think the Column headed "Gene" should read "Target".

10. Table 6. The column headed "Serovar/Name" is inappropriate. Replace with "Serovar/Strain". Then correctly record (using ATCC and NCTC numbers for those strains obtained from ATCC and NCTC and original strain names for others) that actual strain tested. Add an extra row for the field isolates of serovar 5 and ensure that the reference serovar 5 strain or strains that was used is named.

11. Table 6 - serovar 5 results. It is important that the reference serovar 5 strains be separately recorded as noted in the above point. I found it somewhat difficult to believe the number of field serovar 5 isolates that appear to be listed in Table 6. My problem is based on the data shown in Table 1. In Table 1, there are three reference serovar 5 strains listed and 108 field isolates of serovars 1, 2, 5, 12, 15 and non-typeable. I will assume that all three reference strains of serovar 5 are part of the 104 recorded in Table 6. This means that at least 101 field isolates of serovar 5 were tested according to Table 6. Given that Table 1 indicates that the field isolates represented 5 serovars and non-typeable, it means that almost all of the Thai isolates were serovar 5. Is this the case??? Providing the details of the serovars of the field isolates in Table 1 and then separating the details of the reference strains and the field isolates in Table 6 would resolve this problem.

Minor Comments

1. App or A. pleuropneumoniae. In my view, a bacterium that is worthy of study is worthy of being formally and correctly named at each and every occasion in a scientific MS. While virologists are willing to reduce the names of "their" life forms, the same should not apply to bacteria in my view. Please use the normal conventions for all bacterial names - full genus and species at first use and then abbreviated genus name thereafter. The term App should be removed totally from this MS.
2. Line 26. The word "same" is, in my view, superfluous and should be deleted.
3. Line 120. What animal was the source of the blood cells used in the blood agar and what concentration was used.
4. Tables. Each Table should commence on a new page.
5. Line 219. Suggest inserting "only" ahead of "11 complete".
6. Line 256. Insert "the" ahead of "five new"
7. Lines 264-266. Is the sentence starting "Although not encoding intact" based on data generated in this study or is it based on published data? If based on published work, then that work has to be cited. If based on an analysis done in this work, then the text needs to make this clear.

Reviewer #2 (Comments for the Author):

The manuscript titled "Novel DNA markers for detecting *Actinobacillus pleuropneumoniae*" (#Spectrum01311-21) by Srijuntongsiri et al. describes the identification of novel DNA markers specific for *Actinobacillus pleuropneumoniae* using GWS methods and verification of these markers with PCR amplification. The bioinformatics analysis is well-organized and the paper is well-written. These DNA markers would be helpful for development of alternative methods for identification of *A. pleuropneumoniae*. My major concern is that this work is a preliminary study. Further efforts on the development of specific diagnostic methods and functions of target genes might contribute more to the understanding and control of this bacterium

General comments:

1. Line 168, how about the PCR cycle conditions?
2. Line 267, what's the mean of "significantly similarity"?
3. Table 4, why primers for those 5 markers in *apxIVA* gene are not included?
4. Table 5, why fragment *apxIVA-1* shows 51 matches in 23 incomplete *A. pleuropneumoniae* genomes?
5. Table 6, representative results from gel electrophoresis analysis should be displayed, or may be as supplementary material(s).

Staff Comments:

Preparing Revision Guidelines

Please return the manuscript within 60 days; if you cannot complete the modification within this time period, please contact me. If you do not wish to modify the manuscript and prefer to submit it to another journal, please notify me of your decision immediately so that the manuscript may be formally withdrawn from consideration by Microbiology Spectrum.

We thank the Reviewers for their comments, and our response to each one is given below. Line numbers indicate where changes have been made in response to the Reviewer's comments, and these are additionally highlighted in yellow in the marked copy of the revised manuscript.

Reviewer #1:

Major Comments

1. In my view, the title should use the word "identification" in place of "detection".

Response: The title has been changed to “Novel DNA markers for identification of *Actinobacillus pleuropneumoniae*.”

2. Taxonomy. The authors appear not to be aware of the now widely accepted situation that many of the bacterial species identified in this MS as being in the genus *Actinobacillus* are clearly NOT in this genus. This continued allocation of these problematic organisms to the genus *Actinobacillus* when they are clearly not in the genus is only prolonging and intensifying the confusion. It would help all workers if those organisms that are clearly not members of the genus *Actinobacillus* be named at each and occasion with a clear flag that they are not members of the genus e.g. by placing the in-correct genus name in quotation marks ("*Actinobacillus*") or in square brackets ([*Actinobacillus*]). Based on the latest edition of Bergey's Manual (see Blackall, P.J., Turni, C. 2020. *Actinobacillus*. In Bergey's Manual of Systematics of Archaea and Bacteria, Whitman, W.B., Rainey, F., Kämpfer, P., Trujillo, M., Chun, J., DeVos, P., Hedlund, B., Dedysh, S., eds. (John Wiley & Sons), the following [*Actinobacillus*] species featured in this MS are clearly NOT members of the genus *Actinobacillus* and should always have the genus name clearly marked as either "*Actinobacillus*" or [*Actinobacillus*] with a similar approach for the abbreviated genus name "A." or [A.]:-

- a. "A." delphinocola
- b. "A." indolicus
- c. "A." minor
- d. "A." porcinius
- e. "A." rossi
- f. "A." seminis
- g. "A." succinogenes

For the moment, the final resolution of the position of "A." porcitonillarum also remains unclear and that species should also be flagged as well.

Response: The species mentioned above are now marked with [A.] throughout the manuscript. An explanation of [*Actinobacillus*] species has also been added.

The manuscript (lines 126-128) now reads.

“Some [*Actinobacillus*] species such as [*A.*] *indolicus*, [*A.*] *minor*, and [*A.*] *porcinus* are no longer included in the *Actinobacillus* genus but are not yet allocated to a new genus (22). These species are herein described as [*Actinobacillus.*] ”

3. Lack of technical details. At lines 164-172, the text describes the PCRs used in this work. I found the text to be far too light on detail. There are many Taq enzymes supplied by NEB - which one was used? What were the cycling conditions of the assays?

Response: More details on PCR parameters have been added as suggested.

The manuscript (lines 164 – 178) now reads:

“**Genomic DNA purification and PCR amplification**

Genomic DNA of various bacterial species was extracted using a standard DNA purification protocol (29). PCR was performed using Taq DNA polymerase with Standard Taq Buffer (M0273, New England Biolabs, Ipswich, MA, USA) according to the manufacturer’s protocol. Briefly, PCR reactions were prepared to contain final concentrations of 200 μ M dNTPs, 0.2 μ M of each primer (Table 4), 0.025 U/ μ L Taq DNA polymerase, and 1 ng/ μ L of bacterial genomic DNA. Thirty cycles of 95 $^{\circ}$ C for 30 seconds, 60 $^{\circ}$ C for 1 minute, and 68 $^{\circ}$ C for 1 minute were performed using a C1000 Touch PCR Thermal Cycler (Bio-Rad, Hercules, CA, USA). PCR products were visualized by agarose gel electrophoresis followed by ethidium bromide staining. Alternatively, Luna qPCR Master Mix (M3003, New England Biolabs) was used according to the manufacturer’s protocol. Briefly, qPCR reactions were prepared to contain final concentrations of 0.25 μ M of each primer and 1 ng/ μ L of bacterial genomic DNA. Forty-five cycles of 95 $^{\circ}$ C for 15 seconds and 60 $^{\circ}$ C for 30 seconds were performed using a Bio-Rad CFX96 real-time PCR machine. Fluorescence signals indicative of the presence of PCR products were measured.”

4. Field isolates. Based on Table 1, this study included 108 field isolates. The text could be more informative by indicating the number of isolates per serovar (simply place the number in brackets after the serovar and explain via a footnote).

Response: The number of isolates per serovar has been added in brackets to Table 1 as suggested.

5. Table 1. This Table needs considerable refinement.

The column headed "No." is irrelevant and the column should be deleted.

The column headed Species is actually Genus and Species.

The column headed Source/Reference is mostly presenting strain name and reference (ie ATCC 27088 in the first cell in this column is the strain name - the Source is simply ATCC. As well, the columns headed "Strain Name" and "Source/Reference" are largely duplicating each other in terms of the strain name. The column headed "Strain name" should record the ATCC or CCUG

strain name (readers can consult the ATCC and CCUG web sites for the alternative names of each strain) for all strains obtained from CCUG and ATCC. The Source/Reference column should just be showing ATCC or CCUG (and not the strain name). What was the source of the bacteria without a source name e.g. *A. pleuropneumoniae* serovar 7?

The column headed "Number of strains/isolate" is wasting valuable space. Delete this column and simply use a footnote to record the rare exception where more than a single strain/isolate was used. Where the formal type strain of species has been used the T signifying that the strain is the formal species type strain should be in superscript. ATCC 27088^T (where { } shows superscript) is the type strain for *A. pleuropneumoniae*.

Response: Table 1 has been modified as suggested and now reads:

“Table 1. Bacteria used in this study. ATCC, American Type Culture Collection. CCUG, Culture Collection University of Gothenburg. Numbers in brackets indicate the number of isolates. ^T indicates type strain of the species. Species with [*Actinobacillus*] are not officially included in the *Actinobacillus* genus, but have not yet been assigned to a new genus (22).”

Genus and Species	Serovar	Strain Name	Source/Reference*
Actinobacillus pleuropneumoniae	1	ATCC 27088 ^T	ATCC (33)
	2	ATCC 27089	ATCC (33)
	3	ATCC 27090	ATCC (33)
	4	ATCC 33378	ATCC (34)
	5a	ATCC 33377	ATCC (34, 35)
	5b	L20	(34, 35)
	5	ATCC 55454	ATCC
	6	ATCC 33590	ATCC (36)
	7	WF83	(37)
8	405	(38)	

	9	CVJ13261	(39)
	10	D13039	(40)
	11	56153	(41)
	12	8328	Denmark
	13	N-273	(42)
	14	3906	(43)
	15	HS143	(44)
	16	A-85/14	(45)
	17	16287-1	(46)
	18	7311555	(46)
	19	7213384-1	(5)
	1 [2], 2 [1], 5 [100], 12 [1], 15 [1], non-typable [3]	Field isolates from Thailand [108]	This study
Actinobacillus equuli	-	ATCC 9346	ATCC
“Actinobacillus” indolicus	-	CCUG 39029 ^T	CCUG
Actinobacillus lignieresii	-	ATCC 13372	ATCC
	-	CCUG 41384 ^T	CCUG
“Actinobacillus” minor	-	CCUG 38923 ^T	CCUG
“Actinobacillus” porcinus	-	CCUG 38924 ^T	CCUG

“Actinobacillus” rossi	-	ATCC 27072	ATCC
Actinobacillus suis	-	ATCC 15557	ATCC
	-	ATCC 33415 ^T	ATCC
Actinobacillus ureae	-	ATCC 25976	ATCC
Bibersteinia trehalosi	-	ATCC 33367	ATCC
Glaesserella parasuis	-	ATCC 19417	ATCC
	-	Field isolates from Thailand [6]	This study
Haemophilus influenzae	-	ATCC 33391	ATCC
Mannheimia haemolytica	-	ATCC 29696	ATCC
Pasteurella multocida	-	ATCC 43137	ATCC
	-	ATCC BAA-1113	ATCC
Salmonella enterica subsp. enterica	Choleraesuis	ATCC 7001	ATCC
Streptococcus suis	-	ATCC 43765	ATCC

*The Langford laboratory was the source of bacteria (or gDNA) that were not purchased from ATCC or CCUG. The growth and preparation of derived gDNA from these strains was carried out as described previously (5).

6. Confusion on strain names. As I indicated above, there are problems with confusion over strain names in Table 1. This lack of completed understanding has spoiled over into the general text. At lines 280 - 293, there is Discussion about the absence of *xerD* in a strain called NCTC 11383 (note the space - should be used with all NCTC and ATCC strains). The authors appear not to understand that NCTC 11383 is actually K17. Based on Tables 1 and 6, the pure culture of NCTC 11383/ATCC 33377/K17 was used in this study as a live culture and was tested by PCR and was positive for *xerD*.

Response: The discussion on *xerD* has been modified accordingly. The manuscript (lines 307-309) now reads:

“Nonetheless, *xerD*-specific PCR product was observed when genomic DNA from the ATCC 33377 strain was used as template (Table 1 and Table 6), indicating that *xerD* can also serve as a marker for *A. pleuropneumoniae* identification.”

7. Confusion over strains. Given the confusion highlighted above, the authors need to go back into their data and look for other duplications of strains. There may well be additional duplications of strains that are actually the same strain but simply obtained from ATCC or NCTC or other sources.

Response: In Table 2, strain name has been changed to ATCC or CCUG number when these are available. We found no other duplication.

8. Table 3. Again, remove the first column - it is irrelevant.

Response: The first column has been removed as suggested.

9. Table 5. Again, a poorly laid out Table. The column headed "nucleotide sequence" is simply too bulky and makes this table very hard for the reader to assimilate. Move the details in this column to a Supplementary Table. The column headed "Among WGS of Pasteurellaceae" is very confusing. At a guess, the data being shown in the three sub-columns here is actually limited to the similarities in the available incomplete *A. pleuropneumoniae* genomes. The last column is simply a waste of space and can be replaced by a footnote that records whatever was meant to be shown in this column (it is just empty cells at the moment). I think the Column headed "Gene" should read "Target".

Response: Table 5 has been modified as suggested. Table 5 now reads:

Table 5. *A. pleuropneumoniae*-specific DNA marker candidates identified *in silico*. Percent query cover and percent identity after performing MegaBLAST searches against the nr/nt or whole genome sequence (WGS) databases are shown. No similarity between marker candidates and sequences from other species was found by MegaBLAST.

No.	Target	Locus tag in L20 (CP000569)	Predicted function	Length (NTs)	Match to 11 complete A. pleuropneumoniae genomes		Match to incomplete A. pleuropneumoniae genomes		
					% query cover	% identity	Number of matches in 23 incomplete A. pleuropneumoniae genomes	% query cover	% identity
1	apxIVA-1	APL_0998	Toxin	385	100	100	51 (match more than 1 contig in a genome)	19-100	79.43-100
2	apxIVA-2	APL_0998	Toxin	125	100	100	19	38-100	96.8-100
3	apxIVA-3	APL_0998	Toxin	326	100	100	23	96-100	99.08-100

4	apxIVA-4	APL_0998	Toxin	315	100	100	23	100	100
5	apxIVA-5	APL_0998	Toxin	116	100	100	23	100	100
6	eamA	APL_1023	EamA family transporter; DMT family transporter	203	100	100	23	100	99.51-100
7	nusG	APL_1717	Transcription termination/ anti-termination protein	139	100	100	23	100	100
8	sppA	APL_1268	Signal peptide peptidase, protease IV	105	100	100	23	100	100
9	xerD	APL_1542	Site-specific tyrosine recombinase	149	100	100	22 (absent in contigs of ATCC 33377)	100	100
10	ybbN	APL_0080	Co-chaperone YbbN; putative thioredoxin-like protein	127	100	100	23	100	100

11	ycfL	APL_0 125	YcfL family protein; putative periplasmic lipoprotein	101	100	100	23	100	100
12	ychJ	APL_1 658	YchJ family protein, hypothetical protein, SEC-C motif containing	140	100	100	24 (present twice in strain 4226)	100	99.29-100

10. Table 6. The column headed "Serovar/Name" is inappropriate. Replace with "Serovar/Strain". Then correctly record (using ATCC and NCTC numbers for those strains obtained from ATCC and NCTC and original strain names for others) that actual strain tested. Add an extra row for the field isolates of serovar 5 and ensure that the reference serovar 5 strain or strains that was used is named.

Response: Table 6 has been modified as suggested.

11. Table 6 - serovar 5 results. It is important that the reference serovar 5 strains be separately recorded as noted in the above point. I found it somewhat difficult to believe the number of field serovar 5 isolates that appear to be listed in Table 6. My problem is based on the data shown in Table 1. In Table 1, there are three reference serovar 5 strains listed and 108 field isolates of serovars 1, 2, 5, 12, 15 and non-typeable. I will assume that all three reference strains of serovar 5 are part of the 104 recorded in Table 6. This means that at least 101 field isolates of serovar 5 were tested according to Table 6. Given that Table 1 indicates that the field isolates represented 5 serovars and non-typeable, it means that almost all of the Thai isolates were serovar 5. Is this the case??? Providing the details of the serovars of the field isolates in Table 1 and then separating the details of the reference strains and the field isolates in Table 6 would resolve this problem.

Response: The number of strains and isolates have been listed in more detail in Table 6 as suggested. The majority of Thai *A. pleuropneumoniae* isolates recovered from samples sent to the Veterinary Diagnostic Laboratory, Livestock Animal Hospital, Chulalongkorn University are indeed of serovar 5. Table 6 now reads:

**Table 6. Validation of *A. pleuropneumoniae*-specific markers by PCR**. Genomic DNA of various bacterial species/strains was tested for the presence of candidate marker sequences using PCR. ++, PCR product of expected size was present; -, no PCR product present. Numbers in parentheses are expected PCR product sizes in base pairs (bp).

| Species | Serovar | Strain | Number of strains tested | *apxIVA* | | Marker candidate | | | | | | |
|---|---|---|---|---|---|---|---|---|---|---|---|---|
| | | | | 1L-1R (422) | NEST 1L-1R (377) | *eamA* (192) | *nusG* (117) | *sppA* (83) | *xerD* (74) | *ybbN* (58) | *ycfL* (54) | *ychJ* (66) |
| *A. pleuropneumoniae* | 1 | ATCC 27088$^{T}$, 2 field isolates | 3 | ++ | ++ | ++ | ++ | ++ | ++ | ++ | ++ | ++ |
| | 2 | ATCC 27089, 1 field isolate | 2 | ++ | ++ | ++ | ++ | ++ | ++ | ++ | ++ | ++ |
| | 3 | ATCC 27090 | 1 | ++ | ++ | ++ | ++ | ++ | ++ | ++ | ++ | ++ |
| | 4 | ATCC 33378 | 1 | ++ | ++ | ++ | ++ | ++ | ++ | ++ | ++ | ++ |
| | 5 | ATCC 33377, L20, ATCC 55454, 100 field isolates | 103 | ++ | ++ | ++ | ++ | ++ | ++ | ++ | ++ | ++ |
| | 6 | ATCC 33590 | 1 | ++ | ++ | ++ | ++ | ++ | ++ | ++ | ++ | ++ |
| | 7 | WF83 | 1 | ++ | ++ | ++ | ++ | ++ | ++ | ++ | ++ | ++ |
| | 8 | 405 | 1 | ++ | ++ | ++ | ++ | ++ | ++ | ++ | ++ | ++ |
| | 9 | CVJ13261 | 1 | ++ | ++ | ++ | ++ | ++ | ++ | ++ | ++ | ++ |
| | 10 | D13039 | 1 | ++ | ++ | ++ | ++ | ++ | ++ | ++ | ++ | ++ |
| | 11 | 56153 | 1 | ++ | ++ | ++ | ++ | ++ | ++ | ++ | ++ | ++ |

Minor Comments

1. App or *A. pleuropneumoniae*. In my view, a bacterium that is worthy of study is worthy of being formally and correctly named at each and every occasion in a scientific MS. While virologists are willing to reduce the names of "their" life forms, the same should not apply to bacteria in my view. Please use the normal conventions for all bacterial names - full genus and species at first use and then abbreviated genus name thereafter. The term App should be removed totally from this MS.

Response: App has been replaced with *A. pleuropneumoniae* throughout the manuscript as suggested.

2. Line 26. The word "same" is, in my view, superfluous and should be deleted.

Response: The word "same" has been deleted as suggested. The manuscript (lines 23 – 28) now reads:

“Herein, 12 marker candidates highly conserved (99 – 100% identity) among 34 *A. pleuropneumoniae* genomes (covering 13 serovars) were identified to be *A. pleuropneumoniae*-specific *in silico*, as these sequences are distinct from 30 genomes of 13 other *Actinobacillus* and problematic “*Actinobacillus*” species and more than 1700 genomes of other bacteria in the *Pasteurellaceae* family.”

3. Line 120. What animal was the source of the blood cells used in the blood agar and what concentration was used.

Response: 5% sheep red blood cells were used in blood agar, and this information has been incorporated into the manuscript (lines 115-116) which now reads:

“Briefly, clinical samples were cultured on blood agar (containing 5% sheep red blood cells) with a *Staphylococcus aureus* nurse streak and incubated at 37 °C with 5% CO₂.”

4. Tables. Each Table should commence on a new page.

Response: Page breaks have been added before each Table as suggested.

5. Line 219. Suggest inserting "only" ahead of "11 complete".

Response: “Only” has been added as suggested.

The manuscript (lines 226 – 229) now reads:

“Using comparative genome analysis, based on a strict criterion of 100% nucleotide identity across sequences of 100 - 400 nucleotides conserved in only 11 complete *A.*

pleuropneumoniae genomes, 12 sequences were identified as putatively *A. pleuropneumoniae*-specific (Table 5).”

6. Line 256. Insert "the" ahead of "five new"

Response: “the” has been added as suggested.

The manuscript (lines 270 – 272) now reads:

“In *A. pleuropneumoniae* genomes with both *apxIVA* and *apxIV-S*, the five new *apxIVA* marker candidates match to different regions but are still *A. pleuropneumoniae*-specific *in silico* (Fig. 1B, Table 5).”

7. Lines 264-266. Is the sentence starting "Although not encoding intact" based on data generated in this study or is it based on published data? If based on published work, then that work has to be cited. If based on an analysis done in this work, then the text needs to make this clear.

Response: The complete genome sequence, along with ApxIV pseudogene prediction, of *A. lignieresii* strain NCTC 10568 (accession no. NZ_LR134169) is available on NCBI. This information has been added. The manuscript (lines 278 – 281) now reads:

“Although not encoding an intact ApxIV protein (NCBI accession no. NZ_LR134169), the *A. lignieresii* NCTC 10568 genome contains sequences (comprising multiple open reading frames) sharing 73% identity over 71% of the *A. pleuropneumoniae apxIVA* sequence (71% query cover), as determined by BLASTn.”

Reviewer #2 (Comments for the Author):

The manuscript titled "Novel DNA markers for detecting *Actinobacillus pleuropneumoniae*" (#Spectrum01311-21) by Srijuntongsiri et al. describes the identification of novel DNA markers specific for *Actinobacillus pleuropneumoniae* using GWS methods and verification of these markers with PCR amplification. The bioinformatics analysis is well-organized and the paper is well-written. These DNA markers would be helpful for development of alternative methods for identification of *A. pleuropneumoniae*.

Response: The title has been changed to “Novel DNA markers for identification of *Actinobacillus pleuropneumoniae*” to indicate the scope of the present study (see also response to Reviewer 1, major comment 1).

General comments:

1. Line 168, how about the PCR cycle conditions?

Response: More details on PCR parameters have been added as suggested (lines 164 – 178). See also response to Reviewer 1, major comment 1.

2. Line 267, what's the mean of "significantly similarity"?

Response: MegaBLAST software has a threshold score when two sequences are deemed not similar. More clarification has been added.

The manuscript (lines 281 – 285) now reads:

“Five *A. pleuropneumoniae*-specific *apxIVA* marker candidates identified here do not share significant similarity with the *apxIVA*-like sequences in *A. lignieresii*, as determined by default parameters of MegaBLAST search against databases which include three complete and incomplete *A. lignieresii* genomes (Table 5).”

3. Table 4, why primers for those 5 markers in *apxIVA* gene are not included?

Response: We decided not to test markers within *apxIVA* by PCR because the *apxIVA* gene is a proven marker for *A. pleuropneumoniae* as the sequence of the whole *apxIVA* gene, covering all five *apxIVA* markers, is unique to *A. pleuropneumoniae* and numerous *A. pleuropneumoniae* detection assays have already been developed based on this gene.

The manuscript has been edited to include the information above. Manuscript (lines 201 – 203) and now reads:

“As the *apxIVA* gene, whose sequence is unique to *A. pleuropneumoniae*, is a proven *A. pleuropneumoniae*-specific marker (7, 9, 10, 17), we did not perform PCR to validate the five marker candidates within the *apxIVA* gene.”

4. Table 5, why fragment *apxIVA*-1 shows 51 matches in 23 incomplete *A. pleuropneumoniae* genomes?

Response: Each incomplete genome may consist of hundreds of contigs. For each incomplete genome, the fragment *apxIVA*-1 can match with more than 1 contig. For example, *apxIVA*-1 matches to 3 separate contigs in the genome of strain D13039.

More information, i.e. “(match more than 1 contig in a genome)” has been added to Table 5 after 51 matches.

5. Table 6, representative results from gel electrophoresis analysis should be displayed, or may be as supplementary material(s).

Response: Representative gel electrophoresis results have been added to the supplementary material as requested. Manuscript (lines 212 – 213) now reads:

“Representative gel electrophoresis results are also shown in supplementary material.”

November 28, 2021

Dr. Warangkhanha Songsungthong
National Center for Genetic Engineering and Biotechnology
113 Thailand Science Park
Klong Luang, Pathum Thani 12120
Thailand

Re: Spectrum01311-21R1 (Novel DNA markers for identification of *Actinobacillus pleuropneumoniae*)

Dear Dr. Warangkhanha Songsungthong:

Your manuscript has been accepted, and I am forwarding it to the ASM Journals Department for publication. You will be notified when your proofs are ready to be viewed.

Sincerely,

Sandeep Tamber
Editor, Microbiology Spectrum
